# Fast encirclement of an exceptional point for highly efficient and compact chiral mode converters

Xiaoqian Shu[1,6], Aodong Li[1,6], Guangwei Hu [2,3,6], Jian Wang [1✉], Andrea Alù [3,4✉] & Lin Chen [1,5✉]

Exceptional points (EPs) are degeneracies at which two or more eigenvalues and eigenstates of a physical system coalesce. Dynamically encircling EPs by varying the parameters of a non-Hermitian system enables chiral mode switching, that is, the final state of the system upon a closed loop in parameter space depends on the encircling handedness. In conventional schemes, the parametric evolution during the encircling process has to be sufficiently slow to ensure adiabaticity. Here, we show that fast parametric evolution along the parameter space boundary of the system Hamiltonian can relax this constraint. The proposed scheme enables highly efficient transmission and more compact footprint for asymmetric mode converters. We experimentally demonstrate these principles in a 57 μm-long double-coupled silicon waveguide system, enabling chiral mode switching with near-unity transmission efficiency at 1550 nm. This demonstration paves the way towards high-efficiency and highly integrated chiral mode switching for a wide range of practical applications.

[1] Wuhan National Laboratory for Optoelectronics and School of Optical and Electronic Information, Huazhong University of Science and Technology, Wuhan 430074, China. [2] Department of Electrical and Computer Engineering, National University of Singapore, Singapore, Singapore. [3] Photonics Initiative, Advanced Science Research Center, City University of New York, New York, NY 10031, USA. [4] Physics Program, Graduate Center, City University of New York, New York, NY 10016, USA. [5] State Key Laboratory for Mesoscopic Physics, School of Physics, Peking University, Beijing 100871, China. [6]These authors contributed equally: Xiaoqian Shu, Aodong Li, Guangwei Hu. ✉email: jwang@mail.hust.edu.cn; aalu@gc.cuny.edu; chen.lin@mail.hust.edu.cn

In non-Hermitian systems obeying parity-time (PT) symmetry, the eigenvalues and eigenstates simultaneously coalesce at exceptional points (EPs). In particular, the introduction of tailored gain and loss in photonic systems, including microcavities[1–3], coupled waveguides[4–10], gratings[11,12], and photonic crystals[13,14], has recently enabled EPs and their associated non-Hermitian physics to be studied. The unique topological features of EPs, forming a self-intersecting Riemann surface, have given rise to several exotic phenomena, such as loss-induced transmission enhancement[4], enhanced sensitivity[15–18], unidirectional invisibility[19,20], and single-mode lasing[1,21], which in turn have enabled a variety of novel technologies.

Recent works have demonstrated that strong directional response and chiral state evolution can arise owing to non-adiabatic transitions (NATs)[6–11] when EPs are dynamically encircled. Chiral mode switching has been theoretically and experimentally verified by mapping the Hamiltonian parameters onto structural parameters of coupled waveguides, in which encircling EPs in clockwise (CW) or anticlockwise (ACW) direction leads to different output modes. In particular, the final output modes are the same for a specific encircling handedness even if the input modes are different. For conventional chiral mode converters realized based on these principles, the chiral transmission efficiencies are quite low owing to significant energy dissipation for the eigenstates near EPs, with the maximum efficiency experimentally reported so far being 46%[11]. Recent theoretical studies have demonstrated the ability to realize high-efficiency chiral mode transmission by using a discontinuous encircling protocol via Hamiltonian hopping[22]. The physics behind the Hamiltonian hopping is that the eigenstates of the system Hamiltonian converge as its parameters approach infinity, and the hopping is accessible by transitioning between these states. The designed chiral mode converters based on such a discontinuous encircling loop suffer from additional mode crosstalk arising from mode mismatch, resulting in deterioration of the mode purity.

Here, we show fast and adiabatic parametric evolution along the Hamiltonian parameter space boundary, for which the evolution rate of eigenstates is minuscule. At the same time, along this path one of the two eigenstates undergoes extremely low energy dissipation rates. Hence, such an encircling loop including the parameter space boundary is beneficial to construct a compact asymmetric mode converter with high-efficiency chiral mode transmission. The chiral transmission behavior is theoretically predicted and experimentally demonstrated by mapping the required Hamiltonian parameters onto double-coupled silicon waveguides (DSWs) of 57 μm length, in which near-unity efficiency is achieved with −16 dB crosstalk at 1550 nm. Our protocol significantly improves the efficiency of chiral mode conversion compared with the continuous encircling loop[11] and reduces modal crosstalk in contrast to the virtual loop based on Hamiltonian hopping with −12 dB crosstalk[22].

## Results

**Working principle.** To demonstrate chiral mode transmission with fast parametric evolution, we constructed a system comprised of DSWs (Fig. 1a), for which the region between A and B corresponds to the Hamiltonian parameter space boundary. Such a system is fabricated on a silicon-on-isolator (SOI) wafer with a top silicon layer of 220 nm and a SiO₂ buried layer of 2 μm. A 20-nm-thick chromium layer with varied widths is placed on the first silicon waveguide to introduce a position-dependent loss in the system. The output result is TE₁ (TE₀) mode when TE₀ or TE₁ mode is injected into the waveguides from the left (right) port. To understand the chiral mode switching, we firstly performed full-

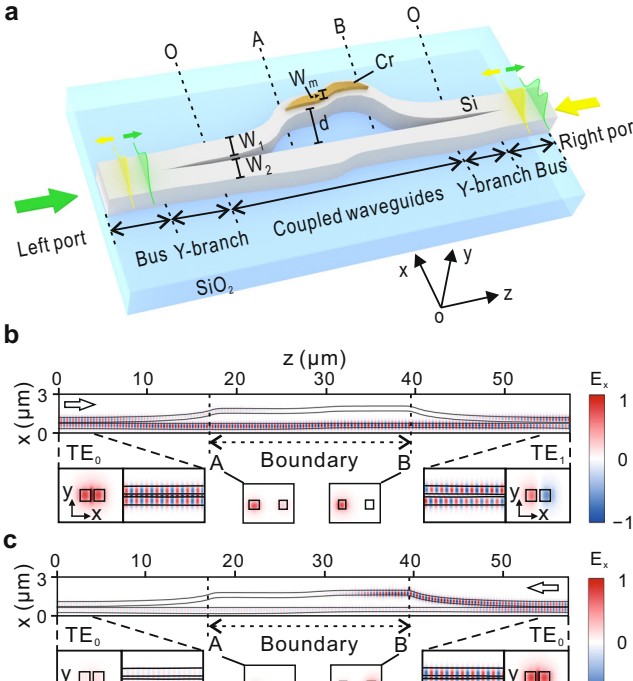

**Fig. 1 DSWs for demonstrating chiral mode transmission with fast parametric evolution. a** Double-coupled silicon waveguides on SOI wafer. The entire device is covered by a 2-μm-thick SiO₂ layer. $W_m$ is the width of the chromium layer, $W_1$ ($W_2$) is the width of the first (second) waveguide, and $d$ is the gap distance between the two waveguides. **b**, **c** Simulated field distributions of $E_x$ at 1550 nm. TE₀ mode inputs from the left port (**b**) and right port (**c**), respectively.

wave simulations using finite-difference time-domain methods (Fig. 1b, c). Specifically, for CW encircling (Fig. 1b), the energy is mainly located in the second waveguide between A and B (associated with the parameter space boundary), where the dominant eigenstate is $[0,1]^T$. The final state is TE₁ with high transmission efficiency. However, for ACW encircling most of the energy is transferred to the first waveguide at B (Fig. 1c), and finally completely dissipated in the waveguide region between A and B. The system state shifts from $[1,0]^T$ to $[0,1]^T$ because of the NAT and finally exits as $[1,1]^T$, corresponding to the TE₀ mode.

To demonstrate these exotic features, such coupled waveguides can be rigorously studied with the evolution equation as

$$\frac{d|\psi\rangle}{dz} = iH|\psi\rangle \quad (1)$$

where the eigenfunction is written as $|\psi(z)\rangle = [a_1(z), a_2(z)]^T$, $a_1(z)$ and $a_2(z)$ are the amplitude of the modes in each waveguide; and the Hamiltonian $H(z)$ can be written as

$$H(z) = \begin{bmatrix} \beta(z) + 2i\gamma(z) & \kappa(z) \\ \kappa(z) & -\beta(z) \end{bmatrix} \quad (2)$$

Here, $z$ is the propagation distance; $\beta(z)$, $\gamma(z)$ and $\kappa(z)$ represent the degree of detuning, loss rate, and coupling strength of the system, respectively, which constitute the parameter spaces for mode evolution. The eigenvalues are $E = i\gamma \pm \sqrt{(\beta + i\gamma)^2 + \kappa^2}$, associated with the normalized eigenstates as $X = \frac{\sqrt{2}}{2}[\sqrt{1 \pm M}, \pm\sqrt{1 \mp M}]^T$, where $M = (\beta + i\gamma)/\sqrt{(\beta + i\gamma)^2 + \kappa^2}$, indicating that the system has an EP at $(\beta/\kappa, \gamma/\kappa) = (0, 1)$. The two eigenvalues are expressed as $E_1$, $E_2$ [Im($E_1$) ≤ Im($E_2$)] and the corresponding eigenstates are

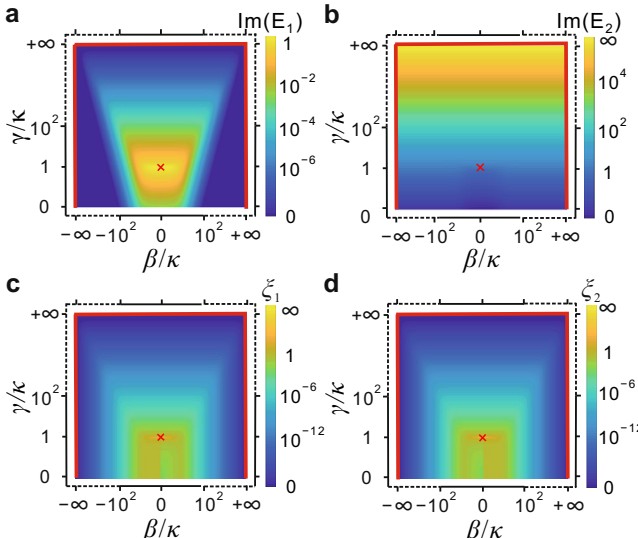

**Fig. 2 Loss and variation rates of eigenstates in the Hamiltonian parameter space. a, b** Imaginary parts of the two eigenvalues, $E_1$, $E_2$, as functions of $\beta/\kappa$ and $\gamma/\kappa$. **c, d** $\xi_1$, $\xi_2$ versus $\beta/\kappa$ and $\gamma/\kappa$. The marked point represents the EP, and the red lines represent the Hamiltonian parameter space boundary.

$X_1$, $X_2$, respectively. Assuming that $H$ remains constant over the distance interval $[z_0, z]$, the final state $\psi(z)$ can be written as

$$|\psi(z)\rangle = c_1(z_0)e^{iE_1(z-z_0)}X_1 + c_2(z_0)e^{iE_2(z-z_0)}X_2 \quad (3)$$

The initial state is $|\psi(z_0)\rangle = c_1(z_0)X_1 + c_2(z_0)X_2$ at $z_0$, with $c_1$ and $c_2$ being arbitrary coefficients. Equation (3) indicates that the variations of phase and amplitude arise from the real and imaginary parts of the eigenvalues, respectively. The imaginary parts of $E_1$ and $E_2$ in parameter space are clearly presented in Fig. 2a, b, indicating that the eigenstate $X_1$ suffers from enhanced loss when approaching the EP and is lossless on the parameter space boundary: $(\beta/\kappa, \gamma/\kappa) \rightarrow (\pm\infty, R)$ and $(\beta/\kappa, \gamma/\kappa) \rightarrow (R, +\infty)$ ($R$: an arbitrary real coefficient), while the loss rate of $X_2$ is nearly proportional to $\gamma/\kappa$.

The Hamiltonian parametric evolution should satisfy the adiabatic approximation to ensure a stable system output state. Otherwise, the output would be mixed with other eigenstates, resulting in low mode purity[12,23]. In the parameter space described by $\vec{s} = (\beta/\kappa, \gamma/\kappa)$, the degree of adiabaticity is $U = |[X_m \cdot (dX_m/d\tau)]/(E_m - E_{m'})|$, and the adiabaticity condition is $U \ll 1$[24], where $X_m$ is the initial eigenstate, $X_{m'}$ is another eigenstate ($m' \neq m$), and $\tau$ is the evolution time that correlates with the propagation distance, $z$, by $d\tau = \kappa dz$. The parametric evolution rate can be written as $ds/d\tau$ ($ds = |d\vec{s}|$), provided that the parameters evolve along the direction $\vec{s}$, yielding $U = \xi_m ds/d\tau$ with $\xi_m = |[X_{m'} \cdot (dX_m/ds)]/(E_m - E_{m'})|$. A smaller $U$ means the adiabaticity condition is more strictly fulfilled, associated with a lower crosstalk of the state. When $\xi_m$ is large, $ds/d\tau$ needs to be small enough to ensure adiabatic evolution. When $\xi_m$ is very small, $ds/d\tau$ can be very large, implying that the parametric evolution is fast, i.e., the device length can be shortened. Figure 2c, d respectively shows $\xi_1$ ($m = 1, m' = 2$), $\xi_2$ ($m = 2, m' = 1$) versus $\beta/\kappa$ and $\gamma/\kappa$ (see Supplementary Note 1 for a detailed derivation). $\xi_1$ and $\xi_2$ increase when they approach the EP, and they are close to zero when approaching the parameter space boundary, where $X_1$ and $X_2$ converge to $[0, 1]^T$ and $[1, 0]^T$, respectively. Consequently, the adiabaticity condition can be guaranteed even if the system experiences fast parametric evolution along the parameter space

boundary, which significantly decreases the necessity of slow parametric evolution and relaxes adiabatic constraints.

It should be emphasized that slow parametric evolution is required to ensure adiabaticity in the previous schemes using a continuous loop near EPs[6,8,11,25]. Meanwhile, the chiral mode transmission efficiency is inevitably low so long as the evolution loop is close to the EPs. High-efficiency chiral mode transmission is expectable if the encircling loop includes the Hamiltonian parameter space boundaries. If the same strategy, i.e., slow evolution in the previous loops[6,8,11,25], is used for a large loop, the resultant device is considerably long. Here we have shown that fast parametric evolution along the parameter space boundaries provides a compact approach to constructing a chiral mode converter, without the sacrifice of transmission efficiency. With the assumption that $U$ is constant during the evolution, the total evolution time for encircling the EP, $\tau_{total}$, can thus be written as $\tau_{total} = \oint \max(\xi_1, \xi_2)ds/U$. For the same $U$, the evolution in the encircling loop including the parameter space boundaries undergoes the comparable $\tau_{total}$, as opposed to the conventional loops near the EPs, but has the advantage of notably boosted transmission efficiency [see Supplementary Fig. 1 for the detailed analysis]. For both loops, we can always increase $\kappa$ to shorten the device length, since $\tau$ is related to $z$ with $d\tau = \kappa dz$.

In other words, a large encircling loop including the parameter space boundary contains the above-mentioned unique features of fast parametric evolution and low loss, which paves the way to realize a short and high-efficiency chiral mode converter. Figure 3 shows the dynamic evolution path of the Hamiltonian for CW (Fig. 3a, b) and ACW (Fig. 3c, d) loops that include the parameter space boundary denoted by blue and red solid lines between A $(-\infty, 0)$ and B $(+\infty, 0)$. The initial state $[1, 1]^T$ at the starting point $(\beta/\kappa, \gamma/\kappa) = (0, 0)$ is located on the upper sheet of the Riemann surface (Fig. 3a). For the whole CW loop, $X_1$ is dominant and it suffers from low loss as the imaginary part of $E_1$ is always zero (Fig. 3b). It is worth emphasizing that $X_2$ is excited since adiabaticity is not strictly fulfilled, but its contribution is small and is further attenuated when we continue the evolution along the

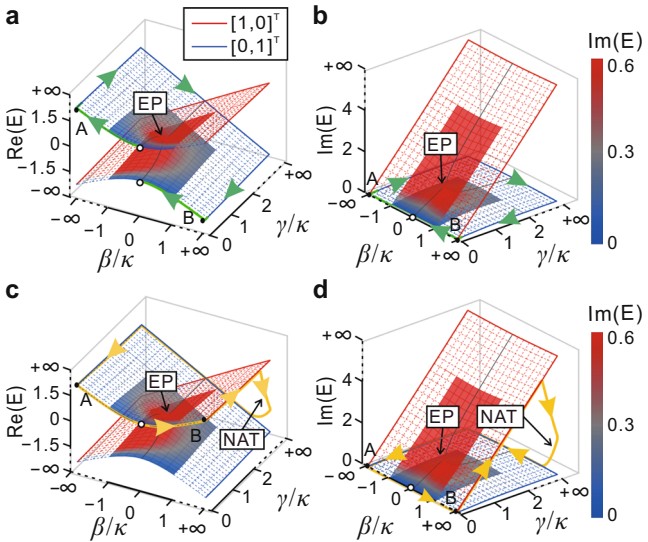

**Fig. 3 System states evolving on the Riemann surfaces. a, b** CW and **c, d** ACW loops around an EP in the Riemann surfaces formed by the real part Re($E$) and the imaginary part Im($E$) of the energy spectra of $H$ as the initial state is $[1, 1]^T$. The empty circles present the start and end of the loops. The dashed meshes present the Hamiltonian parameters extending to infinity. The blue and red solid lines represent the parameter space boundary, associated with the eigenstates as $[0, 1]^T$ and $[1, 0]^T$, respectively.

parameter space boundary. The Hamiltonian finally returns to $(0, 0)$ and the output state is dominated by $X_1 = [1, -1]^T$ on the lower sheet of the Riemann surface. The evolution process and final state for the ACW loop are quite different, as shown in Fig. 3c, d. The initial state $[1, 1]^T$ at the starting point $(\beta/\kappa, \gamma/\kappa) = (0, 0)$ evolves slowly to B, where the dominant eigenstate is $X_2 = [1, 0]^T$ and the other eigenstate $X_1$ is infinitesimal. When the Hamiltonian evolves to A along the parameter space boundary, the dominant eigenstate $X_2$ suffers from high loss and it is completely dissipated. In contrast, $X_1$ is lossless and becomes dominant at A, i.e., a NAT occurs. The final state returns to $[1, 1]^T$ at $(0, 0)$. The completely different output states for opposite encircling handedness indicate a chiral response. It should be noted that the output states for CW and ACW loops are always $[1, -1]^T$ and $[1, 1]^T$ respectively, regardless of the input states. The system evolving along the parameter space boundaries will not cause additional crosstalk for the output state since $\xi_m$ approaches zero. The crosstalk of the output state merely comes from the end of the encircling process, i.e., B to $(0, 0)$ for the CW loop, and A to $(0, 0)$ for the ACW loop. A more detailed dynamical process can be found in Supplementary Notes 2, 3 and Supplementary Figs. 2–4.

**Experimental verification of chiral mode converters.** For our experimental implementation, we use coupled waveguides that render Hamiltonian parameter space boundary easily accessible by suitably selecting the geometrical parameters. Three steps are taken to retrieve the detailed structural parameters. Firstly, based on coupled-mode theory[26], the essential elements in determining the mode evolution are $\beta$, $\gamma$ and $\kappa$, which are controlled by the width difference $\Delta W = W_1 - W_2$, gap distance $d$ and metal width $W_m$. Then a mapping database regarding the structural parameters and Hamiltonian parameters is established. Secondly, the Hamiltonian parameters for the designed encircling loop are selected. Thirdly, based on the mapping database, the corresponding waveguide parameters are determined by the selected Hamiltonian parameters. A part of the mapping database and the detailed structural parameters chosen can be found in Supplementary Note 4 and Supplementary Fig. 5. $TE_0$ and $TE_1$ modes in the silicon bus waveguide stand for $[1, 1]^T$ and $[1, -1]^T$, respectively. Light propagation along the $+z$ ($-z$) direction is associated with CW (ACW) evolution. Figure 4a–c shows the dependence of $\beta$, $\gamma$, and $\kappa$ on $z$ for different wavelengths. The parameter space boundary can be built as follows. Firstly, A (B) can be accessible by letting $\kappa \to 0$ via increased $d$. Secondly, further adjusting $W_m$ and $\Delta W$ makes the Hamiltonian continuously evolve along the parameter space boundary. The parametric evolution trajectory for a shorter wavelength is closer to the parameter space boundary (Fig. 4d). Here it is worth noting that, although the trajectory is not sufficiently close to the boundary, $X_1$ is with ultra-low loss and a fast parametric evolution between A and B is still applicable according to Fig. 2. High-efficiency chiral mode transmission is therefore expected.

A measured scanning electron microscope (SEM) image of the double-coupled waveguides in one of the fabricated samples is shown in Fig. 5a. The zoomed-in image represents the region bounded by the rectangle marked with black lines in Fig. 5a, where the two yellow dashed lines represent the boundaries of the chromium layer. The Cr layer has lower brightness than its surroundings, which can be attributed to two possible factors. In SEM measurement, low primary energy is used to impinge the sample, for the purpose of its protection. More secondary electron on silicon is generated than that on Cr[27]. Besides, the second electron is sensitive to the morphology, and more electron is excited near the sharp corners and boundaries, which induces relatively lower brightness of the Cr layer, in contrast to that near

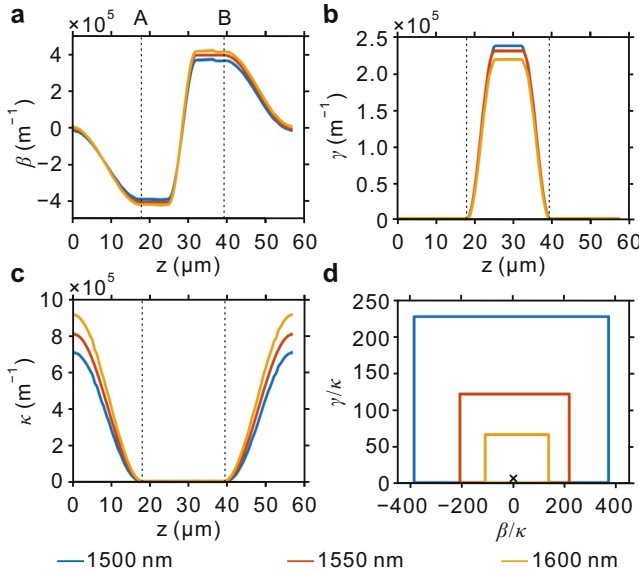

**Fig. 4 Hamiltonian parameters used in the loop. a–c** Dependence of $\beta$ (**a**), $\gamma$ (**b**), and $\kappa$ (**c**) on the spatial coordinate $z$ at 1500, 1550, and 1600 nm, as the interval between A and B denotes the Hamiltonian parameter space boundary. **d** EP-encircling loops at 1500, 1550, and 1600 nm, where the marked point represents the EP.

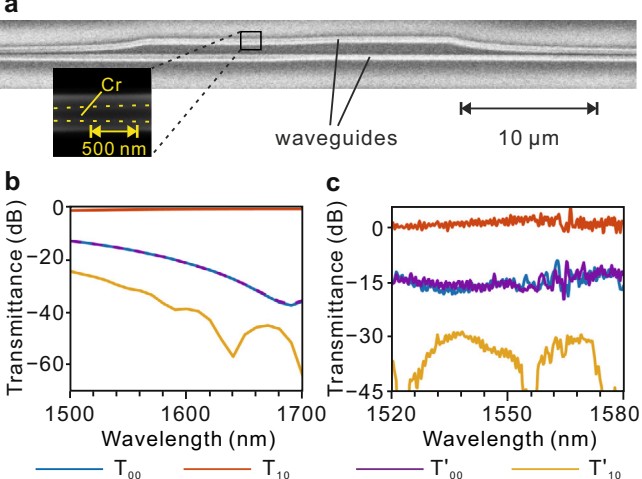

**Fig. 5 Experimental demonstration. a** SEM image of the device. Zoom-in SEM image bounded by the rectangle is marked with black lines. **b** Simulated transmittance spectra for $TE_0$ and $TE_1$ modes at the output port over the wavelength range of 1500–1700 nm. **c** Experimental transmittance spectra for $TE_0$ and $TE_1$ modes at the output port over the wavelength range of 1520–1580 nm. $T_{mn}$ ($T'_{mn}$) represents the transmission efficiency of $TE_m$ mode that outputs from right (left) port when $TE_n$ mode inputs from left (right) port.

the Si-SiO$_2$ interface. Both simulated and experimental results (Fig. 5b, c) demonstrate near-unity transmission efficiency of the $TE_1$ mode at the right port at telecommunication wavelengths when a $TE_0$ mode excites the left port. The mode crosstalk, defined as the ratio of the undesired mode power to the desired one, is $-16$ dB ($-20$ dB) at 1550 nm in the forward (backward) process. Some transmittance dips of $T'_{10}$ are resulted from the fact that the received optical power is lower than the detection limit of the power meter. The discrepancy between simulation and experiment is mainly attributed to imperfect fabrication that

**Table 1 Chiral mode transmission based on encircling EPs.**

| Reference | Efficiency[1] | Crosstalk | Device length | Working wavelength |
|---|---|---|---|---|
| 6 | $10^{-3}$ | −13 dB | 1.25 m | 0.0345–0.0435 m |
| 25 | ~0.04 | −7 dB | 400 mm | 0.0261 m |
| 8 | N.A. | −10 dB | 1 mm | 1250–1700 nm |
| 11 | 0.46 | −10 dB | 69 μm | 1540–1600 nm |
| 22 | ~1 | −12 dB | 120 μm | 1525–1575 nm |
| Our work | ~1 | −16 dB | 57 μm | 1520–1580 nm |

[1]Efficiency is selected as the transmittance for all the different input states and encircling directions are maximum.

may cause different losses from the grating couplers (GCs) and asymmetrical directional couplers (ADCs) in the devices with and without DSWs [see more details in Supplementary Note 6]. The output mode is still $TE_1/TE_0$ mode when a $TE_1$ mode is injected into the waveguide from the left/right port [see the simulated and experimental results in Supplementary Note 7 and Supplementary Figs. 8 and 9].

Finally, although we have only experimentally demonstrated a 57-μm-long asymmetric mode converter, our analytical results reveal the device footprint can be further reduced within the reasonable fabrication requirement. The numerically obtained shortest device has a length of 30 μm, while the transmittance and mode crosstalk does not deteriorate as opposed to the 57-μm-long asymmetric mode converter. If the device length is further reduced, the device performance deteriorates. More details on the analytical results when the device length is varied can be found in Supplementary Note 8 and Supplementary Figs. 10, 11. Based on the current fabrication technology, the fabrication difficulty comes from high-aspect-ratio inductively coupled plasma (ICP) etching for the silicon layer and electron-beam lithography (EBL) alignment precision for Cr layer. For the 57-μm-long asymmetric mode converter, the minimum gap distance between the two silicon waveguides is 85 nm, and the maximum width of Cr layer is 170 nm, much narrower than the silicon waveguide of 400 nm. For the shortest device of 30 μm, the minimum gap distance between the two silicon waveguides is reduced to 50 nm, and the maximum width of Cr layer is 400 nm, with the same width as the silicon waveguide, which significantly challenges our limited fabrication setups.

## Discussion

It is also worth comparing the proposed fast-encircling loop with the previous encircling loops. In the virtual loop based on Hamiltonian hopping[22], a sufficiently long waveguide should be designed to make the system parameters approach infinite points, and the resultant device is 120 μm long with the measured mode crosstalk being of −12 dB at 1550 nm. In contrast, the present scheme does not require a sufficiently long device to let the Hamiltonian parameter rigorously approach the space boundaries during the encircling process. Compared with previous schemes using a continuous encircling loop[6,8,11,25] (Table 1), the present encircling scheme greatly improves the efficiency of chiral mode conversion and experiences a comparable evolution time with the same degree of adiabaticity. Because we have employed a much larger $\kappa$, the resultant device is much shorter than those based on conventional loops near the EPs. Similarly, if $\kappa$ is increased by one order of magnitude in ref. 8, the device length can be shortened, being comparable to our structure [see Supplementary Note 9 and Supplementary Fig. 12 for dynamics analysis]. Our proposal greatly relaxes adiabaticity constraints and can be extended to (anti-) PT-symmetric systems with gain and/or loss in optics, acoustics, and quantum mechanics. Furthermore, asymmetric transformations combined with nonlinear systems are expected to

enable optical isolation[28], and the input-independent output states suggest potential applications in phase-locked laser technology[29]. Our proposed asymmetric transformations with fast parametric evolution are expected to shorten the device length and improve the efficiency, which is of importance to enable compact and high-efficiency optical devices on a chip.

In conclusion, we have shown that non-Hermitian dynamics do not necessarily require slow parametric evolution to ensure adiabaticity. The adiabatic approximation is achievable via fast parametric evolution when operating along the Hamiltonian parameter space boundary, over which one of the eigenstates undergoes extremely low energy dissipation rates. Based on this principle, we have proposed an EP-encircling scheme including the boundary and experimentally demonstrated high-efficiency chiral mode switching in DSWs with only 57 μm length. These results offer a new perspective to studying non-Hermitian dynamics of EP encircling, empowering new opportunities for practical devices and applications that require a smaller footprint and enable higher efficiency compared to conventional schemes.

## Methods

**Fabrications**. The devices were fabricated by a combination of three-step EBL, ICP etching, electron-beam evaporation (EBE), and plasma-enhanced chemical vapor deposition (PECVD). The first-step EBL and EBE were used to form the Au marks on an SOI wafer for alignment. The second-step EBL and ICP were employed to define the waveguide pattern and transfer it onto the SOI wafer. Finally, a chromium layer on top of the silicon waveguide was formed by a third-step EBL with careful alignment, EBE, and lift-off process, which was followed by the PECVD to deposit a 2-μm-thick $SiO_2$ cladding layer to cover the entire device. More fabrication details are given in Supplementary Note 5 and Supplementary Fig. 6.

**Measurements**. GCs and ADCs were placed on both sides of DSWs for transmission measurement. A tunable laser (Santec TSL-710) provides the near-infrared light and its polarization is adjusted by a polarization controller. GC is used to couple the light from the fiber into $TE_0$ mode or decouple the $TE_0$ mode out of the silicon waveguide back into the fiber. The decoupled light will be collected by the optical power meter (PMSII-A), and then is processed by the computer. The optical power in the wavelength range of interest was retrieved by adjusting the wavelength of the tunable laser. see more details on the measurements in Supplementary Note 6 and Supplementary Fig. 7.

## Data availability

All relevant data that support the findings of this study are available from the corresponding author upon reasonable request.

## Code availability

The code that supports the plots within this paper is available from the corresponding authors upon reasonable request.

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

## Acknowledgements

This work is supported by National Natural Science Foundation of China (Grant no. 12074137), National Key Research and Development Project of China (Grant numbers 2018YFB2200200 and 2021YFB2801903), State Key Laboratory of Artificial Micro-structure & Mesoscopic Physics (Peking University), State Key Laboratory of Advanced Technology for Materials Synthesis and Processing (Wuhan University of Technology), and the Air Force Office of Scientific Research MURI program and the Simons Foundation. We thank Pan Li in the Center of Micro-Fabrication and Characterization (CMFC) of WNLO for the support in plasma-enhanced chemical vapor deposition, and the Center for Nanoscale Characterization & Devices (CNCD), WNLO, HUST for the support in SEM measurement.

## Author contributions

X.S., A.L., and L.C. conceived the idea and initiated the work. L.C. guided the project. X.S. and A.L. developed the theoretical framework and performed the numerical simulations. X.S. and J.W. developed the setup and performed the measurements. G.H. and A.A. discussed the results. X.S., A.L., L.C., and A.A. wrote the manuscript and all authors reviewed the manuscript.

## Competing interests

The authors declare no competing interests.
