## [Peer review file · Nature Communications]

REVIEWER COMMENTS

Reviewer #1 (Remarks to the Author):

This manuscript demonstrates on-chip asymmetric mode converters by utilizing the fast parametric evolution in the process encircling exceptional points. The proposed scheme has advantages of high efficiency and compact footprint. In my opinion, the idea of the work is interesting, and the analysis, simulation and experimental components are rather comprehensive. However, I would like the authors to address some important comments/questions before I make the final decision.

1. Table 1 summarizes the comparison between the current work and literature. It looks to me that the performance of Ref. 22 and the current work is similar (120 μm and 57 μm are at the same scale). It will be very helpful if the authors can elaborate more on the novelty and significance of the work. In particular, I wonder if and how the proposed scheme could reduce the footprint down to $\sim 10 \mu\text{m}$, at least from analytical/simulation point of view.
2. It is unclear how the authors came up with the specific waveguide design shown in Fig. S4. More details are needed to explain how the widths of the waveguides and chromium layer, the gap, etc. were designed.
3. The authors could provide better/additional SEM images of the fabricated devices.
 - (1) In Fig. 5a, only the two thin lines (marked by “waveguides”) are the actual waveguides, right? The two wider and bright stripes appear as waveguides on the first look.
 - (2) In Fig. 5b, why does the Cr layer have lower brightness?
 - (3) In Fig. S6b, what does SCWs stand for? Why not showing double-coupled silicon waveguides (DSWs) as stated in the figure caption?
4. Please double check the experimental results presented in Fig. 5d and Fig. S8b. The first curve on the top implies the transmittance exceeding 100%.
5. On page 5, some symbols are missed in the equations.

Reviewer #2 (Remarks to the Author):

This work has shown a non-Hermitian chiral mode converter in a photonic waveguide system. Especially, fast parameter evolution satisfying the adiabatic approximation has been proposed by encircling exceptional points along the boundary of the Hamiltonian parameters, which enables the realization of a

compact chiral converter while maintaining high transmittance and low cross talk. The results show a clear advantage in miniaturization over other similar previous studies. Calculations are high quality and explain the nature of the system. Furthermore, their model has been experimentally verified with a 57- μm long device, which agrees with their theoretical predictions. However, I have a few comments before recommending the publication of this manuscript in Nature Communications.

1) Although one of the main advantages of this work is to reduce the size of the devices, this aspect has been less discussed. I am wondering if there is further room to reduce the current device length (57 μm). The discussion about this can provide a clear prospect of this system. For instance, calculations showing the coupled waveguide length-dependent Hamiltonian parameters may provide the limit of the device size under the current model.

2) Likewise, the relationship between structural parameters (for instance, ΔW , d , W_m , and Chromium length) and the Hamiltonian parameters are not shown. This information may describe the degree of freedom of the device design.

3) Eq. (3) in line 114 indicates Eq. (2)?

4) In Figure 5d, I find a few deeps of T10' intensity, which is not shown in the simulated transmittance spectra. Can you explain this?

Reviewer #3 (Remarks to the Author):

Review report: Manuscript # NCOMMS-21-34305

This manuscript by Shu et al. proposes an efficient EP-encircling process scheme that enables compact and high-performance chiral waveguide devices. The proposed process scheme makes clever use of the parametric space boundary as an encircling-EP loop in order to take advantages of extremely relaxed evolution speed limit as well as extremely high modal purity of the dynamic states with respect to the ideal adiabatic states. Realization in the standard Si-photonics platform results in a chiral mode converter with record high power efficiency and modal purity in a structure with the shortest footprint length ever reported so far. I believe that the proposed scheme is not only original but also provides significant impact on the non-Hermitian physics and its applications. Therefore, I gladly recommend this manuscript for publication in Nature Communications.

I have a few comment that the authors may include in the revised manuscript, as delineated below.

(1) The explanation for the key idea is provided in the second paragraph on page 5 but it is incomplete yet. In this paragraph, high evolution speed $v = |\partial s / \partial t|$ is understood well on the basis of the characteristics $|E_2 - E_1| \rightarrow \infty$ and $\xi_m \approx 0$ along the parametric space boundary. However, total length L of the encircling-EP loop along the parametric space boundary tends to infinity and thereby total evolution time, or device footprint length equivalently, is not necessarily small even with extremely large v . In a complete explanation, the authors may provide a theoretical argument verifying that the adiabatic condition allows v scaling with L or higher, and the total evolution time converges to a certain finite value which can be significantly smaller than that for conventional loops in the moderate size regime.

(2) Another important aspect of the proposed approach is high modal purity but related explanation is not included. Modal purity of a dynamic state is associated with non-adiabatic transition amplitude [ref. 27 in the manuscript] with a factor $|E_2 - E_1|^{-1}$ which tends to 0 along the parametric space boundary. It should be good to include an appropriate explanation on this basis. In addition, the authors may include some comment on the modal purity towards the end of a process over which the parametric condition returns to the initial condition from the parametric space boundary. Note that the final state purity must be determined for this stage, not for steering on the parametric space boundary.

(3) In the definition of factor ξ_m , the additional index n is dummy as there is a constraint $n \neq m$ out of two indices 1 and 2 but n seems to be an independent index in its appearance in the definition. It is good to state this point or use an alternative index-representation scheme in the definition in order to avoid potential confusion. What about using an indexing scheme like " m " (instead of " n ") and say m and m' are binary exclusive indices?

(4) Simple typo: Eq. (3) in line 5 on page 5 should be Eq. (2).

Response to the Reviewers' Comments on the manuscript NCOMMS-21-34305 entitled "Fast
Encirclement of an Exceptional Point for Highly Efficient and Compact Chiral Mode Converters"
submitted to Nature Communications

We thank the editor for handling our manuscript and the reviewers for careful and constructive
opinions. We feel truly encouraged by the shared comments from three reviewers that our work
presents interesting chiral mode converters with high efficiency and compact footprint by fast
encirclement of an exceptional point. We are also very grateful for the minor criticisms from the
reviewers, which have motivated us to significantly improve our manuscript. Below, please find the
point-by-point response in details. We hope our efforts could be appreciated by the reviewers and
firmly believe that our novel findings in this majorly revised manuscript can meet the stringent
requirements and attract general interest of broad readerships of Nature Communications.

**Reviewer #1:**

This manuscript demonstrates on-chip asymmetric mode converters by utilizing the fast parametric
evolution in the process encircling exceptional points. The proposed scheme has advantages of high
efficiency and compact footprint. In my opinion, the idea of the work is interesting, and the analysis,
simulation and experimental components are rather comprehensive. However, I would like the authors
to address some important comments/questions before I make the final decision.

**Response:** We thank the reviewer for the positive comments and endorsements, which have
encouraged us a lot. Below, we have provided the point-by-point response. Especially, we have added
some simulation to elaborate more on the novelty and significance of the work, as compared with Ref.
22.

1. Table 1 summarizes the comparison between the current work and literature. It looks to me that the
performance of Ref. 22 and the current work is similar ($120\ \mu\text{m}$ and $57\ \mu\text{m}$ are at the same scale). It
will be very helpful if the authors can elaborate more on the novelty and significance of the work. In
particular, I wonder if and how the proposed scheme could reduce the footprint down to $\sim 10\ \mu\text{m}$, at
least from analytical/simulation point of view.

**Response:** We are thankful to the reviewer for raising this issue, which has guided us to enrich the
novelty and significance of the present work. We compared with Ref. 22 from different perspectives,
including the physical mechanism and performance of co-designed asymmetric mode converters,
which all show our advantages over the previous one in terms of the footprint and output crosstalk.

- • **Physics:** The asymmetric mode conversion in Ref. 22 is achieved by Hamiltonian hopping on
parameter space boundary, associated with a virtual and discontinuous encircling loop. The
physics behind the Hamiltonian hopping is that the eigenstates of the system Hamiltonian
converge as its parameters approach infinity, and the hopping is accessible by transitioning

between these states. In this newly submitted manuscript, we have employed fast encirclement of
an exceptional point, and the encircling loop, including the Hamiltonian space boundaries, is
enclosed and continuous. In contrast to conventional loops where the parametric evolution must
be sufficiently slow to ensure adiabaticity [A1-A4], we found that such constraint can be relaxed
via fast parametric evolution along the boundary in parameter spaces of the system's Hamiltonian.

- • **Performances:** In practice, infinite points cannot be rigorously reached in Ref. 22, which will
somewhat increase the crosstalk of the output mode. In this case, a sufficiently long asymmetric
waveguide converter should be designed to make the system parameters approach the infinite
points, and the resultant device is 120 μm long with the measured mode crosstalk being of -12 dB
at 1550 nm. In this newly submitted manuscript, we do not require a sufficiently long device to let
the Hamiltonian parameter rigorously approach the space boundaries during the encircling process.
In the original manuscript, we have experimentally presented that an asymmetric mode converter
is 57 μm long with the measured near-unity transmittance of TE_0 - TE_1 and mode crosstalk being of
49 -16 dB at 1550 nm, which has already shown superiority over the previous scheme in Ref. 22 in
terms of the footprint and output crosstalk. In this revision, we have conducted more extensive
computational simulations to check if the device's footprint can be further reduced on the premise
that the performance of the asymmetric mode converter will not deteriorate. Specifically, the
asymmetric mode converter is comprised of three sections of coupled waveguides, denoted by
$L^{(1)}$, $L^{(2)}$, $L^{(3)}$ (the superscript represents the number of the section), as shown in Fig. S10 of
Supplementary Note 8. To find the shortest device, *it is firstly assumed that the transmittance*
*should not be lower than 96%*, which is also the simulated transmittance value for the 57- μm -long
asymmetric mode converter in the experiment. There are a number of combinations of ($L^{(1)}$, $L^{(2)}$,
$L^{(3)}$) to construct the asymmetric mode converter. We have shown in Fig. S11a the minimum
crosstalk versus the device length, L , which clearly shows that the minimum crosstalk is reduced
with the increased device length. The shortest device length can be as small as 30 μm , while the
device performance does not deteriorate as opposed to the 57- μm -long asymmetric mode
converter. It should be noted that, for ease of fabrication, the practical device we aim to
experimentally demonstrate is 57 μm long. Based on the current fabrication technology, the
fabrication difficulty mainly comes from high-aspect-ratio ICP etching for silicon layer, and EBL
alignment precision for Cr layer. For the 57- μm -long asymmetric mode converter, the minimum
gap distance between the two silicon waveguides is 85 nm, and the maximum width of Cr layer is
170 nm, much narrower than that of the silicon waveguide of 400 nm. The associated aspect-ratio
ICP etching and EBL alignment precision is accessible based on the fabrication technology. For
the shortest device of 30 μm , the minimum gap distance between the two silicon waveguides is
reduced to 50 nm, and the maximum width of Cr layer is 400 nm, with the same width as the
silicon waveguide, which we have to admit significantly challenges our limited fabrication setups.

If the device length is further reduced to 10 μm , we obtain a rather different and inferior
performance. The mode field distribution and transmittance are shown in Fig. B1 below, with the
transmittance of 70% (TE_0 mode inputs from the left port), and crosstalk of 1.7 dB (TE_1 mode

inputs from the left port), which means only 41% of desired TE₁ mode is involved in the output
 port.

Figure B1. Simulation results of a 10- μm -long device. (a, b) Simulated field distributions of E_x at
 1550 nm as TE₀ mode inputs from the left (a) and right ports (b), respectively. (c, d) Simulated
 transmittance spectra for TE₀ and TE₁ modes at the output port over the wavelength range of
 1500-1600 nm, when TE₀ mode (c) or TE₁ mode (d) inputs. T_{mn} (T'_{mn}) represents the transmission
 efficiency of TE_m mode that outputs from right (left) port when TE_n mode inputs from left (right) port.
 The geometrical parameters shown in Fig. S10 of Supplementary Note 8 are $d_{\max}^{(1)} = d_{\min}^{(2)} = 0.2 \mu\text{m}$,
 $d_{\max}^{(3)} = d_{\max}^{(2)} = 0.6 \mu\text{m}$, $d_{\min}^{(1)} = d_{\min}^{(3)} = 0.05 \mu\text{m}$, $\Delta W_{\max}^{(1)} = 0.01 \mu\text{m}$, $\Delta W_{\max}^{(3)} = 0.05 \mu\text{m}$, $L^{(1)} = 4 \mu\text{m}$,
 $L^{(2)} = 1 \mu\text{m}$, $L^{(3)} = 5 \mu\text{m}$. The maximum value of W_m equals W_1 . The dependence of d on z in
 sections 1 and 3 follows z^4 . The dependence of d in section 2 and ΔW (W_m) in each section follows
 $\sin z$.

[A1] Doppler, J. *et al.* Dynamically encircling an exceptional point for asymmetric mode switching.
 *Nature* **537**, 76-79, (2016).

[A2] Yoon, J. W. *et al.* Time-asymmetric loop around an exceptional point over the full optical
 communications band. *Nature* **562**, 86-90, (2018).

[A3] Liu, Q. *et al.* Efficient Mode Transfer on a Compact Silicon Chip by Encircling Moving
 Exceptional Points. *Physical Review Letters* **124**, 153903, (2020).

[A4] Zhang, X.-L., Wang, S., Hou, B. & Chan, C. T. Dynamically Encircling Exceptional Points: In
situ Control of Encircling Loops and the Role of the Starting Point. *Physical Review X* **8**, 021066,
(2018).

**Changes made to the paper:**

“The physics behind the Hamiltonian hopping is that the eigenstates of the system Hamiltonian
converge as its parameters approach infinity, and the hopping is accessible by transitioning between
these states.” on page 3, paragraph 1 in the main text.

“Finally, although we have only experimentally demonstrated a 57- μm -long asymmetric mode
converter, our analytical results reveal the device footprint can be further reduced within the reasonable
fabrication requirement. The numerically obtained shortest device has the length of 30 μm , while the
transmittance and mode crosstalk do not deteriorate as opposed to the 57- μm -long asymmetric mode
converter. If the device length is further reduced, the device performance deteriorates. More details on
the analytical results when the device length is varied can be found in Supplementary Note 8. Based on
the current fabrication technology, the fabrication difficulty comes from high-aspect-ratio ICP etching
for silicon layer, and EBL alignment precision for Cr layer. For the 57- μm -long asymmetric mode
converter, the minimum gap distance between the two silicon waveguides is 85 nm, and the maximum
width of Cr layer is 170 nm, much narrower than the silicon waveguide of 400 nm. For the shortest
device of 30 μm , the minimum gap distance between the two silicon waveguides is reduced to 50 nm,
and the maximum width of Cr layer is 400 nm, with the same width as the silicon waveguide, which
significantly challenges our limited fabrication setups.” on page 10, paragraph 2 in the main text.

“In the virtual loop based on Hamiltonian hopping²², a sufficiently long waveguide should be designed
to make the system parameters approach infinite points, and the resultant device is 120 μm long with
the measured mode crosstalk being of -12 dB at 1550 nm. In contrast, the present scheme does not
require a sufficiently long device to let the Hamiltonian parameter rigorously approach the space
boundaries during the encircling process.” on page 10, paragraph 3 in the main text.

“The device is divided into three sections in optimization (Fig. S10a). Four geometrical variables are
the maximum and minimum gap distances, $d_{\max}^{(N)}$, $d_{\min}^{(N)}$, the maximum width difference of the two
waveguides, $\Delta W_{\max}^{(N)}$, and the length, $L^{(N)}$ in each section (the superscript represents the number of
the section, $N=1, 2, 3$). d and ΔW follow monotone variation in each section. In the section 2,
W_m is firstly monotonously increased from zero to the maximum value, and then is kept constant.
Finally, it is monotonously decreased from the maximum value to zero. It is assumed that the
maximum value of W_m is no more than W_1 . Without loss of generality, various functions, including

e^z , z^4 , e^{-z} , $\ln(z)$, and $\sin z$, have been used to describe the dependence of the geometrical
 parameters on z .

To retrieve T_{mm} (T'_{mm}), $t_{uv}^{(N)}$ ($t_{uv}'^{(N)}$) represents the transmission efficiency of TE_u mode that
 outputs from right (left) port when TE_v mode inputs from left (right) port in each section (Fig. S10b).
 The transmittance of the entire device can be written as

$$136 \quad T_{00} = (t_{00}^{(1)}t_{00}^{(2)} + t_{10}^{(1)}t_{01}^{(2)})t_{00}^{(3)} + (t_{00}^{(1)}t_{10}^{(2)} + t_{10}^{(1)}t_{11}^{(2)})t_{01}^{(3)} \quad (S11)$$

$$137 \quad T_{10} = (t_{00}^{(1)}t_{00}^{(2)} + t_{10}^{(1)}t_{01}^{(2)})t_{10}^{(3)} + (t_{00}^{(1)}t_{10}^{(2)} + t_{10}^{(1)}t_{11}^{(2)})t_{11}^{(3)} \quad (S12)$$

$$138 \quad T_{01} = (t_{01}^{(1)}t_{00}^{(2)} + t_{11}^{(1)}t_{01}^{(2)})t_{00}^{(3)} + (t_{01}^{(1)}t_{10}^{(2)} + t_{11}^{(1)}t_{11}^{(2)})t_{01}^{(3)} \quad (S13)$$

$$139 \quad T_{11} = (t_{01}^{(1)}t_{00}^{(2)} + t_{11}^{(1)}t_{01}^{(2)})t_{10}^{(3)} + (t_{01}^{(1)}t_{10}^{(2)} + t_{11}^{(1)}t_{11}^{(2)})t_{11}^{(3)} \quad (S14)$$

$$140 \quad T'_{00} = (t_{00}^{(3)}t_{00}^{(2)} + t_{10}^{(3)}t_{01}^{(2)})t_{00}^{(1)} + (t_{00}^{(3)}t_{10}^{(2)} + t_{10}^{(3)}t_{11}^{(2)})t_{01}^{(1)} \quad (S15)$$

$$141 \quad T'_{10} = (t_{00}^{(3)}t_{00}^{(2)} + t_{10}^{(3)}t_{01}^{(2)})t_{10}^{(1)} + (t_{00}^{(3)}t_{10}^{(2)} + t_{10}^{(3)}t_{11}^{(2)})t_{11}^{(1)} \quad (S16)$$

$$142 \quad T'_{01} = (t_{01}^{(3)}t_{00}^{(2)} + t_{11}^{(3)}t_{01}^{(2)})t_{00}^{(1)} + (t_{01}^{(3)}t_{10}^{(2)} + t_{11}^{(3)}t_{11}^{(2)})t_{01}^{(1)} \quad (S17)$$

$$143 \quad T'_{11} = (t_{01}^{(3)}t_{00}^{(2)} + t_{11}^{(3)}t_{01}^{(2)})t_{10}^{(1)} + (t_{01}^{(3)}t_{10}^{(2)} + t_{11}^{(3)}t_{11}^{(2)})t_{11}^{(1)} \quad (S18)$$

When the efficiency, T_{01} , is set to be 96%, the minimum device length is 30 μm with -16 dB
 minimum crosstalk at 1550 nm (Fig. S11a). The other geometrical parameters are
 $d_{\max}^{(1)} = d_{\min}^{(2)} = 0.43 \mu\text{m}$, $d_{\max}^{(3)} = d_{\max}^{(2)} = 0.64 \mu\text{m}$, $d_{\min}^{(1)} = d_{\min}^{(3)} = 0.05 \mu\text{m}$, $\Delta W_{\max}^{(1)} = 0.09 \mu\text{m}$,
 $\Delta W_{\max}^{(3)} = 0.1 \mu\text{m}$, $L^{(1)} = 14 \mu\text{m}$, $L^{(2)} = 8 \mu\text{m}$, $L^{(3)} = 8 \mu\text{m}$. The maximum value of W_m equals W_1 .
 The dependence of d on z in sections 1 and 3 follows z^4 . The dependence of d in section 2 and ΔW
 (W_m) in each section follows $\sin z$. The FDTD simulated transmittance of the entire device in Fig.
 S11c, d is consistent with the analytical results in Fig. S11a.” in Supplementary Note 8.

**Figure S10. Process of sectional optimization.** (a) The top view of double-coupled silicon
 waveguides used for optimization. (b) The transmission efficiency under TE₀ and TE₁ modes input for
 the three sections.

**Figure S11. Device performance after sectional optimization.** (a) The minimum crosstalk under
 different device lengths after sectional optimization. The star represents -16 dB crosstalk, associated
 with the device length of $30 \mu\text{m}$. (b, c) Simulated transmittance spectra at the output port over the
 wavelength range of $1500\text{-}1600$ nm with TE₀ (b) and TE₁ (c) modes input, while the entire device of 30
 162 μm length in (a), is simulated with FDTD.

2. It is unclear how the authors came up with the specific waveguide design shown in Fig. S4. More
 details are needed to explain how the widths of the waveguides and chromium layer, the gap, etc. were
 designed.

**Response:** We thank the reviewer for pointing out this. The design process can be divided into three
 steps:

- 1) The waveguide parameters including the widths of the waveguides and chromium layer, the gap
 distance between the two coupled waveguides, are mapped into the Hamiltonian parameters based

on the coupled mode equations. Then a mapping database regarding the structural parameters and
 Hamiltonian parameters is established.

2) The Hamiltonian parameters for the designed encircling loop are selected.

3) Based on the mapping database, the corresponding waveguide parameters are determined by the
 selected Hamiltonian parameters.

**Changes made to the paper:**

“Three steps are taken to retrieve the detailed structural parameters. Firstly, based on coupled mode
 theory²⁶, the essential elements in determining the mode evolution are β , γ and κ , which are
 controlled by the width difference $\Delta W=W_1-W_2$, gap distance d and metal width W_m . Then a
 mapping database regarding the structural parameters and Hamiltonian parameters is established.
 Secondly, the Hamiltonian parameters for the designed encircling loop are selected. Thirdly, based on
 the mapping database, the corresponding waveguide parameters are determined by the selected
 Hamiltonian parameters. A part of the mapping database and the detailed structural parameters chosen
 can be found in Supplementary Note 4.” on page 8, paragraph 1 in the main text.

“A part of the mapping database of β , κ , γ with ΔW , d , and W_m is presented in Fig. S5b-d.” in
 Supplementary Note 4.

**Figure S5. Structural parameters of the coupled waveguides.** (a) The top view of double-coupled
 silicon waveguides. (b-d) A part of the mapping database of β , κ , γ with ΔW , d and W_m at 1550
 191 nm. (e-f) The width difference of the two waveguides, $\Delta W = W_1(z) - W_2(z)$, the gap distance, $d(z)$,
 and the width of chromium layer, $W_m(z)$, versus z .

3. The authors could provide better/additional SEM images of the fabricated devices.

**Response:** I apologize that we cannot offer additional SEM images because the sample has been
 already covered by a 2- μm -thick SiO_2 layer. If further measurement is taken, we cannot get better SEM
 images.

(1) In Fig. 5a, only the two thin lines (marked by “waveguides”) are the actual waveguides, right? The
 two wider and bright stripes appear as waveguides on the first look.

**Response:** It is true that only the two thin lines (marked by “waveguides”) are the actual silicon
 waveguides. The two wider and bright stripes are SiO_2 layer. In this revision, we have removed the
 silicon in the upper and lower regions for better SEM image, which has been shown in Fig. 5a.

(2) In Fig. 5b, why does the Cr layer have lower brightness?

**Response:** There are two possible reasons why the Cr layer has lower brightness:

• Figure B2 below schematically shows the relationship between the number of secondary electron,
 δ , and primary energy of electron beam, E_{PE} for Cr and silicon. In our SEM test, the electron
 beam with a low primary energy, E_0 , is used to impinge the sample, for the conservative
 protection of the sample. As a result, the number of the generated secondary electron, δ , on
 silicon is more than that on Cr. This may be the one reason that makes the Cr layer have lower
 brightness [A5].

• The second electron is sensitive to the morphology. More electron is excited near the sharp
 corners and boundaries, associated with enhanced brightness. The flat surface of the Cr layer has
 comparatively lower brightness, in contrast to the region near the Si- SiO_2 interface. This may be
 the other reason.

Figure B2. The number of the generated secondary electron, δ , versus the primary energy of electron

beam, E_{PE} , in SEM.

[A5] Seiler, H. Secondary electron emission in the scanning electron microscope. *Journal of Applied*
*Physics* **54**, R1-R18, (1983).

(3) In Fig. S6b, what does SCWs stand for? Why not showing double-coupled silicon waveguides
(DSWs) as stated in the figure caption?

**Response:** We appreciate the reviewer for pointing out this. Actually, the SCWs in the original
manuscript is the DSWs. In this revision, we have made revisions accordingly.

**Changes made to the paper:**

**Figure 5. Experimental demonstration.** (a) SEM image of the device. Zoom-in SEM image bounded
by the rectangle is marked with black lines. (b) Simulated transmittance spectra for TE_0 and TE_1 modes
at the output port over the wavelength range of 1500-1700 nm. (c) Experimental transmittance spectra
for TE_0 and TE_1 modes at the output port over the wavelength range of 1520-1580 nm. T_{mn} (T'_{mn})
represents the transmission efficiency of TE_m mode that outputs from right (left) port when TE_n mode
inputs from left (right) port.

“The Cr layer has lower brightness than its surroundings, which can be attributed to two possible
factors. In SEM measurement, low primary energy is used to impinge the sample, for the purpose of its
protection. More secondary electron on silicon is generated than that on Cr²⁷. Besides, the second
electron is sensitive to the morphology, and more electron is excited near sharp corners and boundaries,
which induces relatively lower brightness of the Cr layer, in contrast to that near the Si-SiO₂ interface.”
On page 9 in the main text.

**Figure S7. Experimental demonstration.** (a) The experimental configuration. (b) The SEM image of
 the fabricated sample consisting of DSWS, GCs, and ADCs. (c) The SEM image of the control device
 without DSWS.

4. Please double check the experimental results presented in Fig. 5d and Fig. S8b. The first curve on the
 top implies the transmittance exceeding 100%.

**Response:** We are thankful to the reviewer for raising this important issue. As has been stated in the
 last sentence of the first paragraph in Supplementary Note 6, the loss differences in Fig. S7b, c are used
 to extract the transmittance for different TE modes in the DSWS. To measure the transmittance, GCs
 and ADCs are involved in the optical path to input/output the required modes. However, they may
 incur different losses in Fig. S7b, c, as they cannot be exactly the same in fabrication. In our
 measurement, we have recorded the output power in Fig. S7b, c, marked as P_{out1} , and P_{out2} , as the
 input power is the same, marked as P_{in} . The loss coefficients from the GCs, ADCs, and DSWS are
 assumed to be α_{GC} , α_{ADC} , and α_{DSW} . We can thus establish two equations for retrieving the
 transmittance $T (= P_{out1} / P_{out2})$: $P_{out1} = P_{in} \cdot e^{-(\alpha_{GC1} + \alpha_{ADC1})} \cdot e^{-\alpha_{DSW}}$, associated with Fig. S7b, and
 $P_{out2} = P_{in} \cdot e^{-(\alpha_{GC2} + \alpha_{ADC2})}$, associated with Fig. S7c. If the total loss coefficient from GCs and ADCs in
 Fig. S7b, $\alpha_{GC1} + \alpha_{ADC1}$, is smaller than that in Fig. S7c, $\alpha_{GC2} + \alpha_{ADC2}$, the extracted transmittance T
 can be larger than 100%. In the revision, we have reviewed some papers working on silicon
 waveguides, where the same method was adopted to retrieve the mode transmittance, and the measured
 transmittance curves exceed 100% as well [A6-A7].

[A6] Xu, L. *et al.* Ultra-Broadband and Compact Two-Mode Multiplexer Based on
Subwavelength-Grating-Slot-Assisted Adiabatic Coupler for the Silicon-on-Insulator Platform.
*JOURNAL OF LIGHTWAVE TECHNOLOGY* **37**, 5790-5800, (2019).

[A7] Wang, H. *et al.* Compact Silicon Waveguide Mode Converter Employing Dielectric Metasurface
Structure. *Advanced Optical Materials* **7**, 1801191, (2019).

**Changes made to the paper:**

“The discrepancy between simulation and experiment is mainly attributed to imperfect fabrication that
may cause different losses from the GCs and ADCs in the devices with and without DSWs [see more
details in Supplementary Note 6].” on page 9 in the main text.

“The transmittance for TE₀-TE₁ exceeds 100% in some wavelengths. This may come from the
imperfect fabrication that causes different losses from the GCs and ADCs in the devices with and
without DSWs. In our measurement, we have recorded the output power in Fig. S7b, c, marked as
P_{out1} , and P_{out2} , as the input power is the same, marked as P_{in} . The loss coefficients from the GCs,
ADCs, and DSWs are assumed to be α_{GC} , α_{ADC} , and α_{DSW} . We can thus establish two equations for
retrieving the transmittance $T (= P_{out1} / P_{out2})$: $P_{out1} = P_{in} \cdot e^{-(\alpha_{GC1} + \alpha_{ADC1})} \cdot e^{-\alpha_{DSW}}$, associated with Fig. S7b,
and $P_{out2} = P_{in} \cdot e^{-(\alpha_{GC2} + \alpha_{ADC2})}$, associated with Fig. S7c. If the total loss coefficient from GCs and ADCs
in Fig. S7b, $\alpha_{GC1} + \alpha_{ADC1}$, is smaller than that in Fig. S7c, $\alpha_{GC2} + \alpha_{ADC2}$, the extracted transmittance
T can be larger than 100%.” in Supplementary Note 6.

5. On page 5, some symbols are missed in the equations.

**Response:** We are grateful to the reviewer for raising this issue. We have added the missed symbols in
the revision. The content of this part has been revised according to the third reviewer’s comments.

**Changes made to the paper:**

“In the parameter space described by $\vec{s} = (\beta/\kappa, \gamma/\kappa)$, the degree of adiabaticity is
$U = |[X_{m'} \cdot (dX_m/d\tau)] / (E_m - E_{m'})|$, and the adiabaticity condition is $U \ll 1$ ²⁴, where X_m is the initial
eigenstate, $X_{m'}$ is another eigenstate ($m' \neq m$), and τ is the evolution time that correlates with the
propagation distance, z , by $d\tau = \kappa dz$. The parametric evolution rate can be written as $ds/d\tau$
($ds = |d\vec{s}|$), provided that the parameters evolve along the direction \vec{s} , yielding $U = \xi_m ds/d\tau$ with
$\xi_m = |[X_{m'} \cdot (dX_m/ds)] / (E_m - E_{m'})|$.” on page 5, paragraph 2 in the main text.

**Reviewer #2:**

This work has shown a non-Hermitian chiral mode converter in a photonic waveguide system.
Especially, fast parameter evolution satisfying the adiabatic approximation has been proposed by
encircling exceptional points along the boundary of the Hamiltonian parameters, which enables the
realization of a compact chiral converter while maintaining high transmittance and low cross talk. The
results show a clear advantage in miniaturization over other similar previous studies. Calculations are
high quality and explain the nature of the system. Furthermore, their model has been experimentally
verified with a 57-um long device, which agrees with their theoretical predictions. However, I have a
few comments before recommending the publication of this manuscript in Nature Communications.

**Response:** We thank the reviewer for the positive comments and endorsing the quality of our
manuscript, which has encouraged a lot. Below, we have provided the point-by-point response to your
comments to reinforce this work.

1. Although one of the main advantages of this work is to reduce the size of the devices, this aspect has
been less discussed. I am wondering if there is further room to reduce the current device length (57 um).
The discussion about this can provide a clear prospect of this system. For instance, calculations
showing the coupled waveguide length-dependent Hamiltonian parameters may provide the limit of the
device size under the current model.

**Response:** We thank the reviewer for this valuable suggestion. In this revision, we have conducted
more extensive computational simulation to check whether the device's footprint can be further
reduced without the sacrifice of the performance of the asymmetric mode converter. Specifically, the
asymmetric mode converter is comprised of three sections of coupled waveguides, denoted by $L^{(1)}$,
$L^{(2)}$, $L^{(3)}$ (the superscript represents the number of the section), as shown in Fig. S10 of
Supplementary Note 8. To find the shortest device, *it is firstly assumed that the transmittance should*
*not be lower than 96%*, which is also the simulated transmittance value for the 57- μm -long asymmetric
mode converter in the experiment. There are a number of combinations of $(L^{(1)}, L^{(2)}, L^{(3)})$ to
construct the asymmetric mode converter. We have shown in Fig. S11a the minimum crosstalk versus
the device length, L , which clearly shows that the minimum crosstalk is reduced with the increased
device length. The shortest device length can be as small as 30 μm , while the device performance does
not deteriorate as opposed to the 57- μm -long asymmetric mode converter. It should be noted that, for
ease of fabrication, the practical device we aim to experimentally demonstrate is 57 μm long. Based on
the current fabrication technology, the fabrication difficulty comes from high-aspect-ratio ICP etching
for silicon layer, and EBL alignment precision for Cr layer. For the 57- μm -long asymmetric mode
converter, the minimum gap distance between the two silicon waveguides is 85 nm, and the maximum
width of Cr layer is 170 nm, much narrower than that of the silicon waveguide of 400 nm. The
associated aspect-ratio ICP etching and EBL alignment precision is accessible based on the fabrication
technology. For the shortest device of 30 μm , the minimum gap distance between the two silicon
waveguides is reduced to 50 nm, and the maximum width of Cr layer is 400 nm, the same as the width

of the silicon waveguide, which we have to admit significantly challenges our limited fabrication
 setups.

**Changes made to the paper:**

“Finally, although we have only experimentally demonstrated a 57- μm -long asymmetric mode
 converter, our analytical results reveal the device footprint can be further reduced within the reasonable
 fabrication requirement. The numerically obtained shortest device has the length of 30 μm , while the
 transmittance and mode crosstalk do not deteriorate as opposed to the 57- μm -long asymmetric mode
 converter. If the device length is further reduced, the device performance deteriorates. More details on
 the analytical results when the device length is varied can be found in Supplementary Note 8. Based on
 the current fabrication technology, the fabrication difficulty comes from high-aspect-ratio ICP etching
 for silicon layer, and EBL alignment precision for Cr layer. For the 57- μm -long asymmetric mode
 converter, the minimum gap distance between the two silicon waveguides is 85 nm, and the maximum
 width of Cr layer is 170 nm, much narrower than the silicon waveguide of 400 nm. For the shortest
 device of 30 μm , the minimum gap distance between the two silicon waveguides is reduced to 50 nm,
 and the maximum width of Cr layer is 400 nm, with the same width as the silicon waveguide, which
 significantly challenges our limited fabrication setups.” on page 10, paragraph 2 in the main text.

“The device is divided into three sections in optimization (Fig. S10a). Four geometrical variables are
 the maximum and minimum gap distances, $d_{\max}^{(N)}$, $d_{\min}^{(N)}$, the maximum width difference of the two
 waveguides, $\Delta W_{\max}^{(N)}$, and the length, $L^{(N)}$ in each section (the superscript represents the number of
 the section, $N = 1, 2, 3$). d and ΔW follow monotone variation in each section. In the section 2,
 W_m is firstly monotonously increased from zero to the maximum value, and then is kept constant.
 Finally, it is monotonously decreased from the maximum value to zero. It is assumed that the
 maximum value of W_m is no more than W_1 . Without loss of generality, various functions, including
 e^z , z^4 , e^{-z} , $\ln(z)$, and $\sin z$, have been used to describe the dependence of the geometrical
 parameters on z .

To retrieve T_{mn} (T'_{mn}), $t_{uv}^{(N)}$ ($t'_{uv}^{(N)}$) represents the transmission efficiency of TE_u mode that
 outputs from right (left) port when TE_v mode inputs from left (right) port in each section (Fig. S10b).
 The transmittance of the entire device can be written as

$$T_{00} = (t_{00}^{(1)}t_{00}^{(2)} + t_{10}^{(1)}t_{01}^{(2)})t_{00}^{(3)} + (t_{00}^{(1)}t_{10}^{(2)} + t_{10}^{(1)}t_{11}^{(2)})t_{01}^{(3)} \quad (\text{S11})$$

$$T_{10} = (t_{00}^{(1)}t_{00}^{(2)} + t_{10}^{(1)}t_{01}^{(2)})t_{10}^{(3)} + (t_{00}^{(1)}t_{10}^{(2)} + t_{10}^{(1)}t_{11}^{(2)})t_{11}^{(3)} \quad (\text{S12})$$

$$T_{01} = (t_{01}^{(1)}t_{00}^{(2)} + t_{11}^{(1)}t_{01}^{(2)})t_{00}^{(3)} + (t_{01}^{(1)}t_{10}^{(2)} + t_{11}^{(1)}t_{11}^{(2)})t_{01}^{(3)} \quad (\text{S13})$$

$$T_{11} = (t_{01}^{(1)}t_{00}^{(2)} + t_{11}^{(1)}t_{01}^{(2)})t_{10}^{(3)} + (t_{01}^{(1)}t_{10}^{(2)} + t_{11}^{(1)}t_{11}^{(2)})t_{11}^{(3)} \quad (\text{S14})$$

$$T'_{00} = (t_{00}^{(3)} t_{00}^{(2)} + t_{10}^{(3)} t_{01}^{(2)})t_{00}^{(1)} + (t_{00}^{(3)} t_{10}^{(2)} + t_{10}^{(3)} t_{11}^{(2)})t_{01}^{(1)} \quad (\text{S15})$$

$$T'_{10} = (t_{00}^{(3)} t_{00}^{(2)} + t_{10}^{(3)} t_{01}^{(2)})t_{10}^{(1)} + (t_{00}^{(3)} t_{10}^{(2)} + t_{10}^{(3)} t_{11}^{(2)})t_{11}^{(1)} \quad (\text{S16})$$

$$T'_{01} = (t_{01}^{(3)} t_{00}^{(2)} + t_{11}^{(3)} t_{01}^{(2)})t_{00}^{(1)} + (t_{01}^{(3)} t_{10}^{(2)} + t_{11}^{(3)} t_{11}^{(2)})t_{01}^{(1)} \quad (\text{S17})$$

$$T'_{11} = (t_{01}^{(3)} t_{00}^{(2)} + t_{11}^{(3)} t_{01}^{(2)})t_{10}^{(1)} + (t_{01}^{(3)} t_{10}^{(2)} + t_{11}^{(3)} t_{11}^{(2)})t_{11}^{(1)} \quad (\text{S18})$$

When the efficiency, T_{01} , is set to be 96%, the minimum device length is $30 \mu\text{m}$ with -16 dB
 minimum crosstalk at 1550 nm (Fig. S11a). The other geometrical parameters are
 $d_{\text{max}}^{(1)} = d_{\text{min}}^{(2)} = 0.43 \mu\text{m}$, $d_{\text{max}}^{(3)} = d_{\text{max}}^{(2)} = 0.64 \mu\text{m}$, $d_{\text{min}}^{(1)} = d_{\text{min}}^{(3)} = 0.05 \mu\text{m}$, $\Delta W_{\text{max}}^{(1)} = 0.09 \mu\text{m}$,
 $\Delta W_{\text{max}}^{(3)} = 0.1 \mu\text{m}$, $L^{(1)} = 14 \mu\text{m}$, $L^{(2)} = 8 \mu\text{m}$, $L^{(3)} = 8 \mu\text{m}$. The maximum value of W_m equals W_1 .
 The dependence of d on z in section 1 and 3 follows z^4 . The dependence of d in section 2 and ΔW
 (W_m) in each section follows $\sin z$. The FDTD simulated transmittance of the entire device in Fig.
 S11c, d is consistent with the analytical results in Fig. S11a.” in Supplementary Note 8.

**Figure S10. Process of sectional optimization.** (a) The top view of double-coupled silicon
 waveguides used for optimization. (b) The transmission efficiency under TE_0 and TE_1 modes input for
 the three sections.

**Figure S11. Device performance after sectional optimization.** (a) The minimum crosstalk under
 different device lengths after sectional optimization. The star represents -16 dB crosstalk, associated
 with the device length of $30 \mu\text{m}$. (b, c) Simulated transmittance spectra at the output port over the
 wavelength range of $1500\text{-}1600$ nm with TE_0 (b) and TE_1 (c) modes input, while the entire device of 30
 387 μm length in (a), is simulated with FDTD.

2. Likewise, the relationship between structural parameters (for instance, ΔW , d , W_m , and Chromium
 length) and the Hamiltonian parameters are not shown. This information may describe the degree of
 freedom of the device design.

**Response:** We thank the reviewer for raising this issue. In this revision, we have added the relationship
 between ΔW , d , W_m , and the Hamiltonian parameters based on the coupled mode theory. The
 chromium length is not related to the Hamiltonian parameters. We have added the mapping database in
 Supplementary Note 4.

**Changes made to the paper:**

“A part of the mapping database of β , κ , γ with ΔW , d , and W_m is presented in Fig. S5b-d.” in
 Supplementary Note 4.

**Figure S5. Structural parameters of the coupled waveguides.** (a) The top view of double-coupled
 silicon waveguides. (b-d) A part of the mapping database of β , κ , γ with ΔW , d and W_m at 1550
 402 nm. (e-f) The width difference of the two waveguides, $\Delta W = W_1(z) - W_2(z)$, the gap distance, $d(z)$,
 and the width of chromium layer, $W_m(z)$, versus z .

3. Eq. (3) in line 114 indicates Eq. (2)?

**Response:** The reviewer is right on this issue. Because we have added a new equation (1), the updating
 order for equation (2) in the original manuscript is now equation (3) in the revision. We have corrected
 it in the revision.

**Changes made to the paper:**

“Equation (3) indicates that the variations of phase and amplitude arise from the real and imaginary
 parts of the eigenvalues, respectively.” on page 5, paragraph 1 in the main text.

4. In Figure 5d, I find a few deeps of T10' intensity, which is not shown in the simulated transmittance
 spectra. Can you explain this?

**Response:** We are thankful to the reviewer for the valuable comment. The transmittance of T'_{10} is
 very low, compared with other transmittances in principle. As shown in Fig. S7b, c, the grating
 couplers and asymmetrical directional couplers on both sides of the double-coupled silicon waveguides

are involved in the optical path to estimate the transmittance for all output modes in the experimental
measurement. The involved grating couplers and asymmetrical directional couplers will make the
optical power detected by the power meter even lower. In some wavelengths associated with the deeps,
the received optical power is lower than the detection limit of the power meter.

**Changes made to the paper:**

“Some transmittance dips of T'_{10} are resulted from the fact that the received optical power is lower
than the detection limit of the power meter.” on page 9 in the main text.

**Reviewer #3:**

This manuscript by Shu et al. proposes an efficient EP-encircling process scheme that enables compact
and high-performance chiral waveguide devices. The proposed process scheme makes clever use of the
parametric space boundary as an encircling-EP loop in order to take advantages of extremely relaxed
evolution speed limit as well as extremely high modal purity of the dynamic states with respect to the
ideal adiabatic states. Realization in the standard Si-photonics platform results in a chiral mode
converter with record high power efficiency and modal purity in a structure with the shortest footprint
length ever reported so far. I believe that the proposed scheme is not only original but also provides
significant impact on the non-Hermitian physics and its applications. Therefore, I gladly recommend
this manuscript for publication in Nature Communications.

I have a few comment that the authors may include in the revised manuscript, as delineated below.

**Response:** We thank the reviewer for the positive comments and endorsing the novelty of this work,
which has greatly encouraged us. We are also grateful for reviewer's inspiring comments and
suggestions. Below, we have provided the point-by-point response.

1. The explanation for the key idea is provided in the second paragraph on page 5 but it is incomplete
yet. In this paragraph, high evolution speed $V=|\partial s/\partial t|$ is understood well on the basis of the
characteristics $|E_2-E_1|\rightarrow\infty$ and $\xi_m\approx 0$ along the parametric space boundary. However, total length L of
the encircling-EP loop along the parametric space boundary tends to infinity and thereby total evolution
time, or device footprint length equivalently, is not necessarily small even with extremely large V . In a
complete explanation, the authors may provide a theoretical argument verifying that the adiabatic
condition allows V scaling with L or higher, and the total evolution time converges to a certain finite
value which can be significantly smaller than that for conventional loops in the moderate size regime.

**Response:** We thank the reviewer for this valuable suggestion, which is very beneficial to enrich the
key idea. For any loop encircling the EP, we have $L=\bar{v}\tau$ to describe the relationship between the
evolution speed, \bar{v} , and total evolution time, τ , when the total length of the encircling-EP loop, L ,
is fixed. For either our or conventional loop near the EPs, τ can be always adjustable by changing \bar{v} .
If the same strategy, i.e., slow evolution in previous loops, is used for a large loop, the resultant device
length is considerably long. We have shown that fast parametric evolution along the Hamiltonian
parameter boundaries provides a compact approach to constructing a chiral mode converter, without the
sacrifice of the transmission efficiency. In this revision, we have theoretically retrieved τ and
transmission efficiency for the two kinds of loops, and concluded that, with the same degree of
adiabaticity, the evolution in our loop undergoes the comparable τ , as opposed to the conventional
loops near the EPs, but has the advantage of notably boosted transmission efficiency. Because we have
employed a much larger κ , the resultant device is much shorter than those based on conventional

loops near the EPs. If the conventional loop employs a much larger κ , the device length can be
reduced, being comparable to our scheme.

**Changes made to the paper:**

“In the parameter space described by $\vec{s} = (\beta/\kappa, \gamma/\kappa)$, the degree of adiabaticity is
$U = |[X_{m'} \cdot (dX_m/d\tau)]/(E_m - E_{m'})|$, and the adiabaticity condition is $U \ll 1$ ²⁴, where X_m is the initial
eigenstate, $X_{m'}$ is another eigenstate ($m' \neq m$), and τ is the evolution time that correlates with the
propagation distance, z , by $d\tau = \kappa dz$. The parametric evolution rate can be written as $ds/d\tau$
($ds = |d\vec{s}|$), provided that the parameters evolve along the direction \vec{s} , yielding $U = \xi_m ds/d\tau$ with
$\xi_m = |[X_{m'} \cdot (dX_m/ds)]/(E_m - E_{m'})|$. A smaller U means the adiabaticity condition is more strictly
fulfilled, associated with a lower crosstalk of the state.” on page 5, paragraph 2 in the main text.

“It should be emphasized that slow parametric evolution is required to ensure adiabaticity in the
previous schemes using a continuous loop near EPs^{6, 8, 11, 25}. Meanwhile, the chiral mode transmission
efficiency is inevitably low so long as the evolution loop is close to the EPs. High-efficiency chiral
mode transmission is expectable if the encircling loop includes the Hamiltonian parameter boundaries.
If the same strategy, i.e., slow evolution in previous loops^{6, 8, 11, 25}, is used for a large loop, the resultant
device is considerably long. Here we have shown that fast parametric evolution along the Hamiltonian
parameter boundaries provides a compact approach to constructing a chiral mode converter, without the
sacrifice of the transmission efficiency. With the assumption that U is constant during the evolution,
the total evolution time for encircling the EP, τ_{total} , can thus be written as $\tau_{total} = \oint \max(\xi_1, \xi_2) ds / U$.

For the same U , the evolution in the encircling loop including the parameter boundaries undergoes
the comparable τ_{total} , as opposed to the conventional loops near the EPs, but has the advantage of
notably boosted transmission efficiency [see Supplementary Note 1 for the detailed analysis]. For both
loops, we can always increase κ to shorten the device length, since τ is related to z with
$d\tau = \kappa dz$.” on page 6, paragraph 2 in the main text.

**Figure S1. Theoretical analysis with the two encircling loops.** (a-c) An encircling loop including the
 Hamiltonian parameter boundaries, and its total evolution time, τ_{total} , and transmittance for different U .
 (d-f) An encircling loop near the EPs (with a moderate size regime), and its total evolution time, τ_{total} ,
 and transmittance for different U . Here, l_a , l_b , l_c and l_d are used to describe the size of encircling
 loops.

**Figure S12. Dynamical loop near the EP.** (a) The encircling loop at 1550 nm extracted from Ref.
 [A1]. (b, c) The coefficients, $|c_1|$, $|c_2|$, versus the propagation distance, z , for (b) CW and (c) ACW
 loops, when the initial state is $[1,1]^T$ and $\kappa \approx 8500 \text{ m}^{-1}$ (approximately retrieved from the presented
 structure in Ref. [A1]). (d, e) The coefficients, $|c_1|$, $|c_2|$, versus the propagation distance, z , for (d)
 CW and (e) ACW loops, when the initial state is $[1,1]^T$ and $\kappa = 10^5 \text{ m}^{-1}$ (assumed value, one order
 of magnitude larger than the one used in Ref. [A1]).

“The evolution loop used in Ref. [A1] is presented in Fig. S12a. The extracted κ used in the loop is
around 8500 m^{-1} . We can thus retrieve the Hamiltonian parameters and calculate the dynamics of the
loop. The estimated crosstalk is around -10 dB with the length of 1 mm (Fig. S12b, c), well consistent
with the value shown in Ref. [A1]. However, if κ is increased by one order of magnitude to 10^5 m^{-1} ,
the crosstalk is still around -10 dB , but the device length can be shortened to $90 \text{ }\mu\text{m}$.” in
Supplementary Note 9.

2. Another important aspect of the proposed approach is high modal purity but related explanation is
not included. Modal purity of a dynamic state is associated with non-adiabatic transition amplitude [ref.
27 in the manuscript] with a factor $|E_2 - E_1|^{-1}$ which tends to 0 along the parametric space boundary. It
should be good to include an appropriate explanation on this basis. In addition, the authors may include
some comment on the modal purity towards the end of a process over which the parametric condition
returns to the initial condition from the parametric space boundary. Note that the final state purity must
be determined for this stage, not for steering on the parametric space boundary.

**Response:** We fully agree with the reviewer on this point. In this revision, we have added some
presentations to clarify this issue.

**Changes made to the paper:**

“The system evolving along the parameter space boundaries will not cause additional crosstalk for the
output state since ξ_m approaches zero. The crosstalk of the output state merely comes from the end of
the encircling process, i.e., B to (0, 0) for the CW loop, and A to (0, 0) for the ACW loop. More
detailed dynamical process can be found in Supplementary Notes 2 and 3.” on page 7, paragraph 1 in
the main text.

“In addition, it can be seen from Fig. S4a, c that, the system evolving along the parameter space
boundaries will not cause additional crosstalk for the output state. The crosstalk of the output state
merely comes from the end of the encircling process, i.e., B to (0, 0) for the CW loop, and A to (0, 0)
for the ACW loop.” in the last two sentences of the second paragraph of Supplementary Note 3.

3. In the definition of factor ξ_m , the additional index n is dummy as there is a constraint $n \neq m$ out of two
indices 1 and 2 but n seems to be an independent index in its appearance in the definition. It is good to
state this point or use an alternative index-representation scheme in the definition in order to avoid
potential confusion. What about using an indexing scheme like “ m ” (instead of “ n ”) and say m and
m' are binary exclusive indices?

**Response:** We thank the reviewer for this suggestion and have used m and m' to avoid the possible
confusion. In addition, we have emphasized that m should not be equal to m' .

**Changes made to the paper:**

“In the parameter space described by $\vec{s} = (\beta/\kappa, \gamma/\kappa)$, the degree of adiabaticity is
$U = |[X_{m'} \cdot (dX_m/d\tau)]/(E_m - E_{m'})|$, and the adiabaticity condition is $U \ll 1^{24}$, where X_m is the initial
eigenstate, $X_{m'}$ is another eigenstate ($m' \neq m$), and τ is the evolution time that correlates with the
propagation distance, z , by $d\tau = \kappa dz$.” on page 5, paragraph 2 in the main text.

4. Simple typo: Eq. (3) in line 5 on page 5 should be Eq. (2).

**Response:** The reviewer is right on this issue. Because we have added a new equation (1), the updating
order for equation (2) in the original manuscript is now equation (3) in the revision. We have corrected
it in the revision.

**Changes made to the paper:**

“Equation (3) indicates that the variations of phase and amplitude arise from the real and imaginary
parts of the eigenvalues, respectively.” on page 5, paragraph 1 in the main text.

REVIEWERS' COMMENTS

Reviewer #1 (Remarks to the Author):

The authors have successfully addressed all my comments, and thoroughly revised the manuscript. I recommend the acceptance of the manuscript.

Reviewer #2 (Remarks to the Author):

The revised manuscript presents a theoretical and experimental demonstration of a compact non-Hermitian chiral mode convertor with high transmittance and low cross talk. The manuscript provides a comprehensive understanding of the working principle and relevant Hamiltonian parameters of the platform. I believe that the quality of work meets the standard of Nature Communications. I recommend the publication of this manuscript in Nature Communications.

Reviewer #3 (Remarks to the Author):

In the previous round of review, I suggested a revision on two issues:

(1) A more complete explanation on the total evolution time for encircling an EP along the parametric space boundary in comparison with EEP loops in the moderate size regime.

(2) Further explanation on the modal purity towards the end of the process.

In the revised manuscript, the authors treat these issues in a proper descriptions and supporting data. Since I do not have any further comments or suggestions for additional revision, I recommend publication of this revised manuscript as it presently is.